Measuring urban vulnerability to climate change using an integrated approach, assessing climate risks in Beijing

Zhang Mingshun zhangmingshun@bucea.edu.cn 1
Liu Zelu 1
van Dijk Meine Pieter 2
1 Beijing Climate Change Response Research and Education Centre, Beijing University of Civil Engineering and Architecture (BUCEA) , Beijing , China
2 International Institute of Social Studies (ISS), Erasmus University Rotterdam , The Hague , the Netherlands
Santoso Agus
Electronic publication date: 2019 May 30
Publication date: 2019
Volume: 7
Electronic Location ID: e7018
Received 2019 Feb 28; Accepted 2019 Apr 25
Copyright: ©2019 Zhang et al.
Copyright year: 2019
Copyright holder: Zhang et al.
License: This is an open access article distributed under the terms of the Creative Commons Attribution License, which permits unrestricted use, distribution, reproduction and adaptation in any medium and for any purpose provided that it is properly attributed. For attribution, the original author(s), title, publication source (PeerJ) and either DOI or URL of the article must be cited.
License URL: https://creativecommons.org/licenses/by/4.0/

Keywords: Vulnerability assessment, Floods, Adaptation to climate change, Climate change, Heat waves, Beijing, Droughts

Funding: The authors received no funding for this work.

==============================
This study is responding to the recommendation made by IPCC’s fifth Assessment Report on establishing a standard for measuring and reporting climate risk and vulnerability. It exemplifies the assessment of urban vulnerability to climate change by an integrated approach. The results indicate that Beijing is highly exposed to multiple climate threats in the context of global climate change, specifically urban heat waves, urban drainage floods and drought. Vulnerabilities to the climatic threats of heat waves, drainage floods and droughts have increased by 5%–15% during the period of 2008–2016 in Beijing. High vulnerabilities to both heat waves and drainage floods have been observed in the urban downtown area and high vulnerability to droughts have been observed in the outskirts. This vulnerability assessment, which addressed climatic threats, provides a holistic understanding of the susceptibility to climate change that could facilitate adaptation to climate change in the future. The developments of threats like flooding, heat waves and droughts are analyzed separately for 16 districts and an integrated vulnerability index for all of Beijing is provided as well.

Introduction

A vulnerability assessment can be a useful tool for providing information about where we are with regard to the development of the adaptive capacity of a city and the adaptation activities of the population with respect to climate change (Amaru & Chhetri, 2013). The United Nations Framework Convention on Climate Change (UNFCCC) notes that “adaptation emerged as a focus area under the United Nations Framework Convention on Climate Change (UNFCCC) in 2001. However, the attention given to it is still not equal to mitigation with regard to target-setting, financing, and institutional frameworks”. By establishing an explicit long-term adaptation goal in Article 7, the Paris Agreement (UNFCCC, 2015) builds on an international consensus on the need of vulnerability reduction and confirms that adaptation to climate change is a key pillar of UNFCCC. Magnan and Ribera observe as a challenge that climate change adaptation achieves ”equal prioritization with mitigation (given) the relative fuzziness of adaptation as a policy area” (Magnan & Ribera, 2016). Ford adds that “mitigation policy constitutes a response to a clear problem (GHG emissions) and can be measured and tracked using standardized and accepted indicators (e.g., tons of carbon). In contrast, adaptation is difficult to define and track, especially in relation to policy issues like development or disaster risk management” (Ford & Berrang-Ford, 2016). Adaptation is place- and context-specific, with no single approach for reducing risks appropriate across all settings (IPCC, 2014). Vulnerability to climate change can be an accepted indicator for measuring and tracking the success of the adaptation process.

A large and diverse literature has emerged how to assess vulnerability to climate change. The common challenges are to develop robust and credible measures using indicators and indices (known as indicator-based vulnerability assessment (IBVA)) at international, national and local levels (Birkmann, 2007; Birkmann, 2005). Since climate change vulnerability assessment is a complex form of risk assessment, indicators to be used cover components of both geophysical and socio-economic systems (El-Zein & Tonmoy, 2015). Unlike a single system, complex geophysical and socio-economic systems requires large number of indicators for capturing all aspects of vulnerability, and thus all IBVA approaches are facing challenges of selection of necessary indicators, aggregation of indicators into an index, linking indicators to policy implication (Bollin & Hidajat, 2006).

There are two methods for aggregating single indicators into indices in the literature. One is weighting single indicators, and the other is non-weighting single indicators. Weighting means that all single indicators will have different contributions to the integrated index, while non-weighting means that all single indicators have the same contributions to the integrated index. Weights are the measures of importance. There are three arguments for weighting indicators. (a) It is priority-based, and tends to be more realistic than non-weighted indicators. By weighting, priority issues can be addressed and emphasized, and reasonable judgments and decisions can be made. This is very important for the efficient distribution of available resources, and identifying key issues that all stakeholders are concerned about and agree upon. (b) It avoids loss of information during the indicators’ aggregation as much as possible. It is obvious that key indicators will have more weights than others, and aggregated index will have close linkages to the key indicators. The most relevant information will be used in the final index. (c) It is participation-based. The weighting process requires public participation for making judgments. Experts, stakeholders as well as citizens may be involved in the weighting process, and their opinions will have significant influence on the weights of indicators. Since there is no precise mathematical relationship between indicators that ultimately indicate or measure vulnerability, there are limitations in weighting geophysical and socio-economic indicators in measuring vulnerability (El-Zein & Tonmoy, 2015). Weighting indicators make more sense at local level. However, the complexity of issues and the lack of scientific weighting methodologies have been obstructing the application of weighting indicators (Tonmoy, El-Zein & Hinkel, 2014; World Economic Forum, 2001). Environmental indicators can be aggregated by weighting according to their contribution to the environmental issues, which can be determined by scientific research. However, it is very difficult to integrate economic and social indicators by weighting. This is why weighting indicators have not yet been commonly used in the aggregation of social and economic indicators. In our research, the Analytical Hierarchy Process (AHP) has been applied, together with expert judgment for integrating weighted indicators into the index of vulnerability. We find that this method is useful for cities like Beijing, where there is a lack of data, where the situation may differ in different locations and where it is impossible to use a large numbers of indicators for measuring vulnerability to climate change.

Two questions remain in measuring vulnerability. The first concerns “the identification of appropriate reference points from which to assess whether we are successfully enhancing adaptive capacity, strengthening resilience and reducing vulnerability to climate change” (Article 7, para 2, the Paris Agreement). Reference points in emission reduction target-setting have been identified by the Kyoto Protocol and Paris Conference. However setting a reference point for adaptation requires much more and diverse information. The second question concerns the review processes for assessing vulnerability and thus measuring progress on adaptation commitments. It ”will need to balance robustness and comparability of units or indicators that capture key aspects of vulnerability and adaptive capacity with being contextually appropriate” (Magnan & Ribera, 2016). “Transparent and consistent decision-making on climate financing will require clarity on how adaptation intersects with broader development and risk reduction efforts, and thus what constitutes a progression beyond previous efforts” (Article 9, para 3, the Paris Agreement).

To assess vulnerability, it is crucial to define vulnerability. The most authoritative and widely quoted definition of vulnerability in the context of climate change is developed by the Fourth Assessment Report of the Intergovernmental Panel on Climate Change (IPCC) (IPCC, 2007). “Vulnerability is the degree to which a system is susceptible to, and unable to cope with, adverse effects of climate change, including climate variability and extremes. Vulnerability is a function of the character, magnitude, and rate of climate change and variation to which a system is exposed, its sensitivity, and its adaptive capacity”. The IPCC definition has significant limitations. Most critically the three components of exposure, sensitivity and adaptive capacity have not been clearly defined and thus the challenge is to develop indicators for measuring the three components mentioned above. Patt observes that “The IPCC concept in general has proven difficult to operationalize in practice” (Patt et al., 2009). There is considerable debate in the literature as to what constitutes adaptive capacity and how it might be recognized (Yohe & Tol, 2002; Brooks, Adger & Kelly, 2005; Vincent, 2004).

A city is a complex system composed of multiple sub-systems that interact in various and often crucial ways (Van Dijk & Zhang, 2005). Vulnerability related to climate change corresponds with risks of all sub-systems in climate change. However, if measuring vulnerability wants to take all dimensions of vulnerability into account, a considerable number of indicators for measuring of vulnerability is required. The challenge is to deal with two controversial issues. One, the growing complexity of the definition of urban vulnerability related to climate change and secondly, the practical demand for simplicity of information concerning vulnerability. There is no consensus how many indicators are needed for effective measurement of urban vulnerability. The point is to identify the essential elements that could capture the relevant information on the total system.

Main challenges for development of vulnerability assessment include (1) lack of data availability or gaps in the data, (2) integrating different sources of information (e.g., peer-reviewed and grey literature, quantitative data with qualitative information and expert judgment), (3) development and use of a unified methodology (e.g., common climate scenarios, metrics), (4) involvement of stakeholders, (5) collection of information is expensive in terms of time and resources and (6) development of effective indicators (European Environment Agency, 2018). This article addresses the mentioned-above challenges of (1)–(4).

The concept of vulnerability is complex and a one dimensional approach could not meet the needs of vulnerability assessment. Hence an integrated approach is necessary. This article is concerned with creating an urban vulnerability index (UVI) following a step-by-step approach in the context of urban China and intends to help policy-makers at the city and district levels to get insight in the factors determining vulnerability by measuring the progress achieved in adaptation to climate change in the urban areas. Our strategy is first to develop an urban vulnerability index, and secondly to measure empirically the UVI in Beijing, the capital city of China and its major districts. It is the first time such an assessment of vulnerability to climate change is taking place in Beijing, although there are various natural disaster risk assessments reported that use two single indicators (economic damages and mortality) without weighting. The methods developed in this research can become a useful tool for other emerging cities as well.

UVI Development: Theory and Data Collection Method

The major research question is in which way we can measure urban vulnerability in Beijing. Subquestions concern the role of floods, the role of heat waves and the role of droughts. Kropp et al. (2009) developed heat wave indicator within the scope of a climate change vulnerability study for North Rhine Westphalia. However, this indicator does not cover the three components of exposure, sensitivity and adaptive capacity. In our research, we propose that vulnerability could be assessed by the dimensions of exposure, sensitivity and adaptive capacity. Exposure is determined solemnly by climatic threats or issues (e.g., heat wave, urban flooding, drought, etc.). The sensitivity is determined by characteristics of sectors, systems and group of people that are directly or indirectly influenced by climatic threats. The adaptive capacity of a system is determined by the system’s capacity to adjust to climatic threats including the ability to learn from experience or information and hence to reduce its sensitivity. Both exposure and sensitivity lead to more vulnerability, while increased adaptive capacity diminishes vulnerability. Figure 1 shows the procedures we developed to guide the assessment of vulnerability. We separate the procedure to do the assessment from the methods used to come up with the necessary data.

Figure 1 Procedures for the assessment and methodology for data collection.

Climate change has different meaning for different cities and different cities face different priority climate themes. In this study, Impact Oriented Monitoring (IOM), together with stakeholders’ participation is applied to identify key climatic threats or priority climate themes that Beijing is facing. IOM is based on historical data of direct economic loss of climatic events, in particular the data concerning the last 12 years in the Beijing Annual Natural Disaster Report. Following the identification of priority climate themes, a one day workshop was organized by the Beijing Climate Change Response Research and Education Centre, of the Beijing University of Civil engineering and Architecture for allowing expert judgment and stakeholders’ participation for identifying vulnerable sectors and groups. Title of this workshop was Vulnerable Sector and Groups: How Beijing Adapt to Climate Change and it took place on 14 April 2017. 30 participants attended the workshop. Among the 30 participants who are all senior experts, 24 (two from each sector) are from sectors like water supply and sewage, waste management, public health, energy supply, urban green and biodiversity, forest and ecosystem, tourism, nature conservation, information and communication, insurance, transportation and education and six are experts recommended by China National Expert Committee on Climate Change Adaptation (CECCCA). Participants were selected on criteria like (1) at least 5 years working experiences in natural disaster governance or climate adaptation, (2) good knowledge of indicator development and index processing, (3) have participated in development of its sectoral strategies and policies on climate change adaptation. Before making a judgment participants listened to a presentation of this research and were discussing vulnerability issues allowing an intensive exchange of ideas between stakeholders and experts.

A two-steps approach was applied for the preliminarily selection of individual indicators. The first step is to do a literature review selecting individual indicators for climatic threats related vulnerable systems/sectors and vulnerable groups/individuals. By the literature reviews, we have selected preliminary 26 individual indicators for assessing urban vulnerability to the threat of floods, 32 indicators for assessing urban vulnerability to the threat of heat waves and 29 individual indicators for assessing vulnerability to the threat of drought. All those preliminary selected 87 indicators, together with a brief description of each indicator were distributed to the participants before the workshop. The second step is to adopt an expert approach and achieve stakeholders’ participation as described above. The experts and stakeholders present their own understanding and comments and finally they make their own scoring for each of indicators, based on the criteria: (1) the indicators are closely related to the issue, (2) they are independent of one another, (3) relevant in the context of Beijing, (4) measurable and (5) the data are available. Based on the sum of the scores of each indicators, the final 23 indicators were selected for measuring urban vulnerability.

A frequently mentioned challenge in indicator programmes (Van Dijk, 2014) is how the index’s inputs should be weighted. In most of international indicator programmes, nominally all inputs into an index receive equal weight. No indicator gets more weight than any other. An environmental sustainability index programme implemented jointly by the World Economic Forum’s global Leaders for Tomorrow Environment Task Force, the Yale Center for Environmental Law and Policy, and the Columbia University Center for International Earth Science Information Network (CIESIN) has also addressed the issue of weighting indicators (World Economic Forum, 2001). However the result is not acceptable for two reasons. One is that weights identified by different experts from different countries are very likely to differ significantly. This is because the situation is different from one country to another. Another reason for getting poor results is probably that vulnerability is a complicated issue and may lead to confusion among different experts. The reason for opting for a weighting approach for the aggregation of the selected indicators are twofold. One to allow cross-urban comparisons. Most Chinese cities are quite similar in terms of climatic threats, economic growth, social progress, institutional capability and governance. Therefore a scientifically acceptable determination of the weights should be achieved. Secondly we used the Analytical Hierarchy Process (AHP) method, together with consultation of experts. One of the main advantages of the AHP is that it allows experts to compare two factors by using the same criteria and scoring rule. As the identification with the same criteria and scoring rule is quite simple, statistical results can easily be obtained at an acceptable level. As an example, Fig. 2 shows how AHP is applied to aggregate weighted indicators into urban sensibility index that is one component of urban vulnerability for the threat of floods.

Figure 2 AHP framework for measuring urban sensibility for the threat of floods.

As shown in Fig. 2, the AHP objective (first hierarchy) is to measure urban sensibility for the threat of floods under three principles (second hierarchy) of (1) relevance, (2) scientifically sound and (3) applicable to users by using weighted individual indicators (third hierarchy) of annual rainstorm hours (indicator (1), percentage of paved area (indicator (2) and risk of secondary disasters caused by heavy precipitation (indicator (3). Relevance is explained as directly related to the issue, based on a known linkage between the indicator and sensibility and sensitive to changes in the conditions of interest. Scientifically sound is explained as unbiased and representative of the conditions of concern, scientifically credible, so that they cannot be easily challenged in terms of their reliability or validity, based on data of a known and acceptable quality and consistent and comparable over time and space. Applicable to users is explained as based on data which are available at an acceptable cost-benefit ratio, easily understood and applicable by potential users, acceptable to stakeholders and available soon after the event or period to which it relates (so that policy decisions are not delayed). Experts and stakeholders compare the three individual indicators against each of the three principles by using the AHP scoring rule in Table 1.

Likewise, the urban exposure index and the urban adaptive capacity index can be calculated by weighted indicators and the urban vulnerability index can be weighted by giving weight to urban sensibility, urban exposure and urban adaptive capacity.

The composite UVI combines numbers of individual indicators measured in different units into a single number between 0 and 1. Kropp et al. (2009) uses fuzzy logic techniques for this combination in order to account for a quantification of uncertainties. Kropp notes that “these techniques allow for gradual instead of binary allocation of variable values to classes”. For example in a region where life expectancy of 25 years old is classified as low and life expectancy of 85 years old is classified as high. A practical value in between these two thresholds is then allocated to the class ‘life expectancy’ according to a linear function that could for example result in the number ‘0.5’ for the value 55 years old.

For applying fuzzy logic techniques, we developed the methodology for standardizing data aiming at generating the value of each of the indicators in a range from 0–1.

Standardized data is calculated by:

I i = (X i − B i)/(A i − B i ) (if I i is a positive indicator. i = 1, 2, 3, …, n)

I i = (B i − X i )/(B i − A i) (if I i is a negative indicator, i = 1, 2, 3, …, n).

Where:

Ii: & Standardized version of an indicator I i. It varies from 0 to 1. Xi: & Real value of an indicator Ii;

Ai: & Theoretically max. value of a positive indicator Ii and theoretically min. value of a negative indicator I i.

Bi: & Theoretically min. value of a positive indicator Ii and theoretically max. value of a negative indicator Ii.

Beijing has a population of 22 million and covers an area of 16,400 km2. The city of Beijing is divided into four areas with 16 districts. Forced by the availability of statistical data, collected mainly by the district statistical bureaus, the research unit is the district, as presented in Table 2.

Table 1 Values of comparison between two factors.

Description	Value of A/B	
1. as important as B	1	
2. between 1 and 3	2	
3. slightly more important than B	3	
4. between 3 and 5	4	
5. really more important than B	5	
6. between 5 and 7	6	
7. much more important than B	7	
8. between 7 and 9	8	
9. very much more important than B	9	

Empirical Results

Priority climate schemes

Based on the data of the past 12 years on economic damages of main climatic events (2005–2016), IOM is used to identify priority climatic threats in Beijing. Owing to the fact that those data are not constant and they are not available every year and, in certain years, data is estimated by the relevant municipal department, IOM result is verified and corrected by key stakeholders.

Table 2 Four areas and 16 districts under this research.

4 areas	Main function	16 districts	
Urban core area (city centre)	downtown and location of central government	2 districts: Dongcheng and Xicheng	
Urban extended area	Main developed area	4 districts: Haidian, Chaoyang, Fengtai and Shijingshan	
Urban new development area	Areas for new development	5 districts: Fangshan, Shunyi, Tongzhou, Changping and Daxing	
Ecological conserving area	Limited development and providing ecological services to the city	5 districts: Mentougou, Huairou, Pinggu, Miyun and Yanqing	

The stakeholders are related to the main vulnerable sectors: urban water supply, communication and information, transportation, energy supply, sewage and drainage, solid waste collection and treatment, health, insurance, urban green and biodiversity, food production and supply, buildings and build-up area, governance and management, poverty reduction and management and tourism. The preliminary IOM on priority climatic threats in Beijing is presented in Fig. 3, which shows that the priority climatic threats in Beijing are urban drainage floods, high temperature and heat waves and water scarcity and drought. Therefore the main sources for adapting to climate change in Beijing are suggested to allocate means to increase adaptive capacities of responding to heavy rain and drainage floods, high temperature and heat waves and to solve the consequences of water scarcity after droughts. Other climatic threats such as heavy snow and frost, windstorm, wildfires, may be also included in the urban adaption action plan by allocating certain resources, e.g., finance and services.

Figure 3 Priority climatic threats in Beijing.

The priority climatic threats in Beijing are urban drainage floods, high temperature and heat waves and water scarcity and drought.

Vulnerable sectors and groups in each of the priority climate schemes

For identifying vulnerable sectors and groups, a one-day workshop that has been described in the ‘UVI Development: Theory and Data Collection Method’ section was organized. At the end of the workshop day, each participant made his or her own selection of vulnerable sectors and groups. Based on the selection of each participant, the research team determines the aggregate average score, and the final result is based on the aggregate average score, which is presented in Tables 3 and 4.

Table 3 Vulnerable sectors and climatic threats (score: 1–10, high score, high vulnerability).

	Heatwave	Floods	Drought	Average	
Water supply	7.5	4.4	7.0	6.3	
Sewage	4.7	8.9	2.6	5.4	
Waste management	7.3	8.1	1.8	5.7	
Heath	8.4	7.3	6.5	7.4	
Insurance	4.4	7.9	6.8	6.4	
Energy supply	8.8	4.2	5.2	6.1	
Build-up areas	3.5	7.2	2.8	4.5	
Transportation	6.2	7.8	3.3	5.8	
Forest and ecosystem	3.8	3.7	8.0	6.3	
Tourism	7.7	6.8	2.8	5.4	
Statistical Analysis	
Mean	6.2	6.6	4.7	5.9	
Standard deviation (SD)	1.98	1.84	2.26	0.77	
Coefficient of variation (SD/Mean ×100%)	31.8%	27.8%	48.4%	13.0%	

Table 4 Vulnerable groups and climatic threats (score: 1–10, high score, high vulnerability).

	Elderly	Sick	Disabled	Child	Low-income	Ethnic minorities	Outdoor workers	
Heatwave	8.8	8.2	6.8	7.3	7.7	2.5	9.3	
Floods	4.5	5.2	7.4	5.7	6.9	2.0	8.3	
Drought	3.7	7.4	3.8	3.2	6.8	2.8	3.2	
Average	5.7	6.9	6.0	5.4	7.1	2.5	6.9	
Notes.

(Data source: primary data).

Table 3 shows the most vulnerable sectors are water supply, waste management, health, energy supply and tourism in the case of high temperature and heat waves, and sewage, waste management, health, insurance, build-up area, transportation and tourism in the case of drainage and floods, and water supply, insurance, health and forest and ecosystem in the case of water scarcity and drought. Aggregate average scores in Table 3 show that there is no significant difference among most sectors in vulnerability to climate change, since the coefficient of variation (C.V.) is 13.0% that is less than 15%. Significant difference exists in each of the climatic threats, since CV is 31.8% in the case of heatwave, 27.8% in the case of floods and 48.4% in the case of drought. The research result suggests the municipality of Beijing should take an integrated approach to climate adaptation, instead of sectoral actions.

Table 4 provides that there is no significant difference of vulnerability among different groups and different climatic threats, except ethnic minorities that have no difference with majorities in Beijing in terms of living standards and lifestyles. Therefore, it is suggested that adaptation action in Beijing will be targeting vulnerable groups of elderly, sick, disabled, children and, in particular low-income and outdoor workers.

UVI in each of priority climate threats

An urban vulnerability index (UVI) that measures vulnerability to climate changes is a crucial part of the adaptation capacity building process. An UVI is an assessment, information and monitoring tool. As an assessment tool, it helps identify risks and vulnerabilities. As an information tool, it can better instruct the design of early-warning systems and adaptation action plan as well as raise awareness and communicate about vulnerability and risks. As a monitoring tool, it can identify how well a city has responded and recovered to disasters and shocks and whether the targets have been met. Hence, an UVI plays a role during the whole adaptation capacity building process. By applying the three-dimension framework presented in Fig. 2, together with literature reviews and expert approach, the composite UVI is presented in the Table 5.

Table 5 Composite UVI in different climatic threats in Beijing.

Urban vulnerability index in the thread of floods (UVIf)
UVIf = 0.3 Is + 0.3 Ie – 0.4Ia	Sensibility Index (Is) (0.30)	I1: Annual rainstorm hours (0.45)
I2: Percentage of paved area (0.35)
I3: Risk of secondary disasters caused by heavy precipitation (0.20)	
	Exposure Index (Ie) (0.30)	I4: Percentage of population above 65 years old (0.63)
I5: Population density (0.37)	
	Adaptive capacity (Ia) (0.40)	I6: Governance capacity (0.35)
I7: Capacity and quality of drainage infrastructure (0.40)
I8: Capacity of early warning (0.25)	
Urban vulnerability index in the thread of heatwave (UVIh)
UVIh = 0.3 Is + 0.3 Ie –0.4Ia	Sensibility Index (Is) (0.30)	I1: Percentage of high-rise building (0.15)
I2: Numbers of days&nights of heatwaves (0.50)
I3: Percentage of urban green area (0.35)	
	Exposure Index (Ie) (0.30)	I4: Percentage of population above 65 years old (0.23)
I5: Percentage of low income (0.31)
I6: Percentage of outdoor workers (0.46)	
	Adaptive capacity (Ia) (0.40)	I7: Governance capacity (0.55)
I8: Capacity of early warning (0.45)	
Urban vulnerability index in the thread of drought (UVId)
UVId = 0.3 Is + 0.3 Ie –0.4Ia	Sensibility Index (Is) (0.30)	I1: Average annual precipitation (0.55)
I2: Days of daily precipitation of less than 0.1 mm (0.45)	
	Exposure Index (Ie) (0.30)
Adaptive capacity (Ia) (0.40)	I3: Biodiversity (0.30)
I4: Percentage of population above 65 years old (0.33)
I5: Share of agricultural GDP in total GDP (0.37)	
	Adaptive capacity (Ia) (0.40)	I6: Governance capacity (0.55)
I7: Capacity of early warning (0.45)	
Notes.

The values in brackets are the weights.

Figure 4 presents data of vulnerability for the climatic threat of drainage floods in the 16 districts of Beijing. It shows that the vulnerability in 2016 increases significantly (T-test result for the long-term trend: T = 24.82 > T(α = 0.05, n − 2 = 7) = 2.365) by 5%–10% compared to 2008, except the district of Shunyi, where the Beijing Capital International Airport is located and where its vulnerability decreased in the past 10 years, due to the high level drainage infrastructure, which has been developed in the airport area and industrial parks near the airport. High vulnerability to the threat of drainage of floods is observed in the urban core areas (downtown) of two districts of Dongcheng and Xicheng, due to the factors of high percentage of paved areas (high sensibility), high population density and high percentage of population above 65 years (high exposure) and old or very old urban drainage system (lower adaptive capacity) that was built 40 years ago.

Figure 4 Vulnerability in the climatic thread of drainage floods.

The vulnerability for the climatic threat of drainage floods in the 16 districts of Beijing shows that the vulnerability in 2016 increases significantly by 5%–10% compared to 2008, except the district of Shunyi, where the Beijing Capital International Airport is located and where its vulnerability decreased in the past 10 years, due to the high level drainage infrastructure.

Data of vulnerability for the climatic threat of high temperature and heat waves is presented in Fig. 5, which shows that the vulnerability in 2016 increases significantly (T-test result for the long-term trend: T = 10.28 > T(α = 0.05, n − 2 = 7) = 2.365) by 10%–15% in all 16 districts compared to 2008. High vulnerability to heat waves was observed in the urban core areas of districts of Dongcheng and Xicheng, due to high percentage of high-rise building and limited green areas (high sensibility), high percentage of population above 65 years old and relative high percentage of outdoor workers (high exposure). Lower vulnerability in heat waves observed in the district of Yangqing, which is the core ecological conservation area with high green area, less population density and less high-rise buildings. Most districts are in middle vulnerability with value between 0.4–0.6. There are relative low vulnerability in districts of urban new development area and in the ecological conservation area of districts of Huairou, Pinggu and Miyun. These districts have low exposure and low sensibility. In practice, these districts are the main ecological conservation and recreation areas in Beijing.

Figure 5 Vulnerability in the climatic thread of high temperature and heat waves.

The vulnerability in 2016 increases significantly by 10%–15% in all 16 districts compared to 2008. High vulnerability to heat waves was observed in the urban core areas of districts of Dongcheng and Xicheng, due to high percentage of high-rise building and limited green areas (high sensibility), high percentage of population above 65 years old and relative high percentage of outdoor workers (high exposure).

Figure 6 shows data of vulnerability for the climatic threat of drought in the 16 districts of Beijing. Figure 6 clearly indicates that vulnerability to drought in 2016 increased significantly (T-test result for the long-term trend: T = 12.90 > T(α = 0.05, n − 2 = 7) = 2.365) by about 10% compared to 2008 in the districts of urban new development area and particularly in the four districts of urban ecological conservation area, but decreases by about 10% in the urban core areas and urban extended areas. The climatic threat of drought mainly influences the urban ecosystem and the sectors of agriculture and tourism, due to high sensibility and exposure in those areas and its impact in the urban downtown areas is limited, due to lower sensibility and exposure.

Figure 6 Vulnerability in the climatic thread of drought.

The vulnerability in 2016 increases significantly by about 10% compared to 2008 in the districts of urban new development area and particularly in the four districts of urban ecological conservation area, but decreases by about 10% in the urban core areas and urban extended area.

Beijing integrated vulnerability index is calculated by the average of the sum of vulnerabilities of climatic threats of floods, heat waves and drought. Figure 7 presents that the integrated vulnerability in the 16 districts increased significantly (T-test result for the long-term trend: T = 22.69 > T(α = 0.05, n − 2 = 7) = 2.365). The UVI in the case of heat waves is highest and it is obviously increasing during the period of 2008 –2016, given to the factors of increasing sensibility (e.g., increasing of numbers of day and night of heat waves annually, limited green space, and increasing of high-rise buildings) and of exposure (e.g., increasing of population density and population above 65 years old, increasing of outdoor workers). Although adaptive capacity in all 16 districts is increasing, the increase is not enough to overcome the increase in both sensibility and exposure. The integrated UVI in the case of drought is almost stabilized, due to that fact that this UVI decreases in the urban core areas and increases in the urban outskirt of ecological conservation areas. The integrated UVI in the case of floods increases slowly, due to the fact that the increase of both sensibility and exposure increases the UVI and the improvement of urban drainage services and governance capacity decreases the UVI.

Figure 7 Integrated vulnerability to climate change in Beijing.

The integrated vulnerability in the 16 districts increased significantly. The UVI in the case of heat waves is highest and it is obviously increasing during the period of 2008–2016, given to the factors of increasing sensibility and of exposure.

Discussion

The vulnerability assessment has dominated climate change adaptation programmes at various levels of community, city, region, country and the world as a whole (Hinkel, 2011; Sheridan & Kalkstein, 2004; Ford et al., 2015). All assessments are developed based on the definitions of vulnerability and this article is based on the IPCC definition. The IPCC definition defines three components of sensibility, exposure and adaptive capacity of vulnerability. The advantages of the IPCC definition are that it is easy to communicate and results can be compared. The limitation is that the three components lack policy relevance. For linking the assessment to adaptation policies and actions, multi-dimensional vulnerability has been defined and multi-dimensional vulnerability encompasses physical, economic, social, environmental and institutional features (Birkmann, 2005). It is obvious that a multi-dimensional vulnerability assessment needs more indicators and data than the three-dimensional model developed in this article. From definition to practical assessment, the most important challenge facing vulnerability assessments is how to contextualize a definition for the local level, community or district level and even street level. This article started from the 87 indicators selected from literature and, by contextualization, finally 23 indicators are selected for assessing vulnerability to climate change. We argue that the three-dimensional model requires limited indicators and data and thus it is better fitting with cities and communities where data is lacking.

The majority of vulnerability assessments focuses on international, national and regional levels and it is necessary to downscale to the community level for supporting local policy framing and adaptation actions (Birkmann, 2005; Lesnikowski et al., 2015; Adger, 2006). In this article, the focus was on the city and district levels. We go beyond the IPCC framing of exposure, sensitivity and adaptive capacity by developing indicators for them. We have communicated this research results with local stakeholders and the feedback shows that vulnerability assessment on a grid scale is much needed, since it provides crucial information for policy makers and stakeholders to take local actions. We argue that, for improving policy relevance and intensifying linkages of the assessment and local action, urban vulnerability assessments need to downscale to the street level. In this regard, combining IBVA with GIS is called for in the future, since GIS could provide data on very small scale, e.g., on scale of one km2.

In this research the Impact Oriented Monitoring (IOM) together with stakeholders’ participation is proven effective to define priority climatic threats. Applying IOM requires collection of high-quality historical data. IOM could provide direct effects of climate disaster and thus it help to communicate with policy makers and stakeholders and helps to understand climate threats that a city may face. By applying IOM, priority climate threats have been identified and thus we capture the key features of vulnerability. Hence it provides valuable and reliable information on vulnerability to difference climatic threats at the city and district level.

The selection of primary indicators is based on a literature study and the reduction and weighting of indicators were the result of a combined experts’ opinions, stakeholders’ participation and using AHP. There are limitations in aggregating weighted indicators using the additive approach that is mainly based on Multiple Attribute Utility Theory (MAUT) (Tonmoy, El-Zein & Hinkel, 2014). Indicators should be independent and there should be a precise mathematical relationship between indicators. In the literature, most indicator-based vulnerability assessments use MAUT for aggregation of indicator despite the fact that IBVA rarely satisfies MAUT. In our study, we have introduced the criterion of indicator independence in the preliminary selection of indicators. We argue that MAUT is applicable to aggregation of indicators, since all IBVA in vulnerability studies are empirical. However, one must be very careful to explain and communicate the IBVA results, since IBVA provides information more on variations of vulnerability to climate change as indicated in Figs. 4 to 7. IBVA hardly provides information on the absolute value of vulnerability.

The results generated by this research indicate that Beijing is highly exposed to multiple climate threats in the context of global climate change, specifically urban high temperature and heat waves, heavy rain and urban drainage floods and reduced precipitation and drought. This vulnerability assessment, which addressed priority climatic threats in Beijing, provides a holistic understanding of the susceptibility to climate change that could be a guidance for facilitating future adaptation.

Conclusions

The following key conclusions could help explain the vulnerability to climate change in Beijing.

1. The IOM result depicted in Fig. 2 shows that priority climatic threats are urban drainage floods, high temperature and heat waves and water scarcity and drought in Beijing. Therefore the main source for adapting to climate change in Beijing is increasing the adaptive capacity to be able to respond to heavy rains and drainage floods, high temperature and heat waves and water scarcity and drought. The IOM results revealed that the most harmful climatic threat was historically urban floods in Beijing. However this is changing recently and high temperature and heat waves are going to be the most harmful climatic threat as indicated in Fig. 6.

2. Vulnerability to high temperature and heat waves in 2016 increased significantly (T-test result for the long-term trend) by 10%–15% in all 16 districts of Beijing compared to 2008. High vulnerability to heat waves is observed in the urban core areas of the districts Dongcheng and Xicheng, due to high sensibility and high exposure in these two districts.

3. Vulnerability to heavy rain and drainage floods in 2016 increases significantly (T-test result for the long-term trend) by 5%–10% as compared to 2008, except the district of Shunyi, where the Beijing Capital International Airport is located and where its vulnerability decreases in the past 10 years. High vulnerability to heavy rain and drainage floods is observed in the urban core areas (downtown) of two districts of Dongcheng and Xicheng.

4. Vulnerability to drought increased significantly (T-test result for the long-term trend) by about 10% in the districts of urban new development area and particularly in the four districts of urban ecological conservation area, but decreased by about 10% in the districts of urban core areas and urban extended areas.

While no measure of such a complex phenomenon can be perfect, we suggest further research in this field. Firstly, it is necessary to operationalize international understanding of vulnerability within local contexts. Local communities are all in different stages of preparedness to adapt to climate change. They need to understand locally vulnerability, and should have the tools to identify the influence of local governmental policies on local vulnerability through a process of bottom-up changes. For supporting local understanding and assessment of vulnerability, it is suggested that future IPCC report could attach a technical guideline on vulnerability assessment that can be directly used in or adapted to local conditions. Contextualization of international understanding and approaches can inform local adaptation policy and help incorporating climate futures in planning, and thus provide support to achieving SDG targets on climate change and urbanization.

Secondly, a number of refinements for the selection of single indicators and their aggregation need to be undertaken to deepen local understanding of vulnerability. The score of UVI highly depends on what indicators have been used, how much weight have been assigned to the various factors, and what kind of vulnerability rule has been set up. It is obvious that there are a variety of value judgments and significant uncertainties about the assured causality. It is necessary to deepen further studies on those issues.

Thirdly, to be useful, the UVI advanced by this research need to be applied in more cities. Applying the UVI in more Chinese cities, and modifying the UVI accordingly are needed. Finally, a major investment in data collection is clearly called for. Both availability and data quality are crucial to a successful measure of vulnerability.

Supplemental Information

Supplemental Information 1 Dataset for Figures 4-7

Raw data sheet: floods, fig.-4 Data calculated by using the method in table 5 shows the vulnerability for the climatic threat of drainage floods in the 16 districts of Beijing. It shows that the vulnerability for the climatic threat of drainage floods in 2016 increases by 5%-10% compared to 2008, except the Shunyi district.

Raw data sheet: heatwave, fig.-5 Data calculated by using the method in table 5 shows the vulnerability for the climatic threat of heat waves in the 16 districts of Beijing. It shows that the vulnerability in 2016 increases by 10%–15% in all 16 districts of Beijing, compared to 2008.

Raw data sheet: drought, fig.-6 Data calculated by using the method in table 5 shows the vulnerability for the climatic threat of drought in the 16 districts of Beijing. It clearly indicates that vulnerability to drought in 2016 increased by about 10% compared to 2008 in the districts of urban new development area and particularly in the four districts of urban ecological conservation area, but decreases by about 10% in the urban core areas and urban extended areas.

Raw data sheet: integrated, fig.-7 Beijing integrated vulnerability index is calculated by the average of the sum of vulnerabilities of climatic threats of floods, heat waves and drought. Fig 7 presents the integrated vulnerability in the 16 districts of Beijing.

Click here for additional data file.

Additional Information and Declarations

Competing Interests

Author Contributions

Data Availability

The authors declare there are no competing interests.

Mingshun Zhang conceived and designed the experiments, performed the experiments, analyzed the data, contributed reagents/materials/analysis tools, authored or reviewed drafts of the paper, approved the final draft.

Zelu Liu conceived and designed the experiments, performed the experiments, contributed reagents/materials/analysis tools, prepared figures and/or tables, approved the final draft.

Meine Pieter van Dijk analyzed the data, authored or reviewed drafts of the paper, approved the final draft.

The following information was supplied regarding data availability:

Data are available as a Supplemental File.

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
