# Peer review of "Measuring urban vulnerability to climate change using an integrated approach, assessing climate risks in Beijing"

_PeerJ, doi:10.7717/peerj.7018_

## Round 0.1 · original submission · Minor Revisions

The revised manuscript was sent back to the two reviewers that previously assessed the submitted version. The two reviewers are satisfied with the authors’ revision. However, based on my evaluation the manuscript still requires some revision. From my comments below, there are three important points that the authors need to address:

1. Statistical significance.
2. Information about the workshop that formed the expert judgment
3. Editorial issues (grammar, typos)

L16: ABSTR ACT
L75: “no a precise” – delete “a”
L78: “Weighting indicators more sense at local level.” – grammatically incorrect

L170 (and others): “has been organized” -> “was organized”. Is there a reference or a website for the workshop? If so, please add the weblink around L170, to show who attended, the title of the workshop, when it was held, etc. This seems to be a necessary information for the readers to appreciate the validity of the method which relies on expert judgment.

L248-250: “For example in a region where …. 85 years old is classified as high” – this sentence is not complete. What is it about the region that the authors are trying to say?
L274: “Given to the fact” -> “Owing to the fact”
L275: “those data” is plural, so use “are” and “they”
L296: This appears to be the same workshop indicated in L170. So please shorten this paragraph by referring to the earlier section and provide a reference to the workshop.
L340: “thread” should be “threat”? There are several other instances of this typo. Please do a global search and replace.
L345: “observed” should be “is observed”
L356: “the Fig 5” should be just “Fig 5”
L362: “Most of districts” -> “Most districts”
L365: “these those” -> “these”
L391: “but these increasing” - > “the increase”
L391: “the increasing of both” -> “the increase in both”
L394-395: Please check this sentence. It is hard to understand.
L408: “relevancy” -> “relevance”
L425: “at very small square” -> “on a grid scale” or “at a very small spatial scale”? also L428-429.
L447: “preliminary the selection” delete “the”
L462: “show” -> “shows”
L463: “in Beijing” is repetitive. Delete.
L471: “observed” -> “is observed”

L341, 356, 373, e.g., L474: “increases by 5%-10% during 2008-2018” - state what this is relative to. These seem to be referring to trends over the period. Are the quoted values per year? Also L479-481 and make this clear elsewhere. Are these increases statistically significant? Have the authors conducted statistical significance test? Please state this clearly in the manuscript.

L311, etc.: “no significant difference” – How is significance measured here?
L477: “observed” -> “is observed”
L485: “adopt” -> “adapt to”? otherwise “adopt climatic challenges” doesn’t make much sense.

Reviewer 1 ·

Basic reporting

good

Experimental design

good

Validity of the findings

good

Additional comments

I am very happy that the authors have addressed my concerns point by point precisely. No further suggestions come from my side. Therefore, I would like to recommend this manuscript to be published.

Reviewer 2 ·

Basic reporting

I congratulate the authors of the paper as the revised manuscript has improved significantly in terms of quality and clarity.

Experimental design

Method section is now well described and the use of IOM has been clarified.

Validity of the findings

Authors did not aim to validate their findings (same as many vulnerability studies). In fact it is difficult to validate vulnerability as it is an intrinsic quality of a system and not often easily measurable. I was surprised to know that this is the first climate change vulnerability study conducted in an important city like Beijing.

Additional comments

I congratulate you for modifying this paper. Well done!

---

## Round 0.2 · Minor Revisions

The authors are thanked for the revision which resulted in a better readability. But there are remaining issues below (see attached PDF for the complete comments).

Editor’s previous comment:
L341 (L341-346), 356(360-361), 373(376-378), e.g., L474(476-478): “increases by 5%-10% during 2008-2018” - state what this is relative to. These seem to be referring to trends over the period. Are the quoted values per year? Also L479-481(480-481) and make this clear elsewhere. Are these increases statistically significant? Have the authors conducted statistical significance test? Please state this clearly in the manuscript.

Author’s response:
All these increases are the value in 2016 as comparing to the value of year 2008 (baseline). Authors have not conducted statistical test, since the results are ONLY for showing development tendency. Statistical test (e.g. T-test) requires certain numbers of observations. For example, we could do a t-test for the means for the first four years and the next five years (the change is in 2012). However, we have only nine observations!

Editor’s response: Thank you for clarifying that these are based on 2016 relative to 2008. Differences between two points could be just due to random fluctuations, not a long-term tendency. It could be inferred visually, but the tendency itself (e.g., linear trend) can have a statistical significance calculated. If those linear trends shown in the figures are statistically significant, then the change in 2016 from 2008 would be significant as well, thus it would indicate a real signal such as climate change.

Suggested method. The statistical significance of the slope of a best-fit line can be evaluated using the t-distribution. The statistic value for a linear regression model y=Beta1*x+Beta0 is: t=(Beta1 – Beta10)/standard error, where Beta1 is the slope of the fitted line, Beta0 is the y-axis transect, and Beta10 can be set as 0 (i.e., testing against null hypothesis of no trend) and the standard error is: standard deviation of y / sqrt(sum(x-mean(x))^2). The rejection region for say 0.05 level would be t > t(alpha=0.05,n-2) where n is the number of sample (in this case 9). This can be found in standard statistics text books (e.g., Devore J. L., Probability and Statistics for Engineering and the Sciences, p. 498).

Now it may be challenging to indicate this in every line of the figures, so the authors may just comment in the main text on a few selected cases. It would be much better though if the statistical significance can be shown in each of the figures by e.g., showing those time series with statistically insignificant slope in thinner lines. I strongly encourage the authors to implement this as it would significantly increase the impact of this paper.

---

## Round 0.3 · accepted · Accept

While I have accepted the manuscript, there are a couple of minor edits that need to be done.

Please, could you make it clear that the t-test result is for the long-term trend. So simply add 'long-term trend' in those added statements, like this:
"significantly (T-test result for the long-term trend:"

Regarding the text edits, previously there were double 'of'. You only need to delete one of them. So it needs to read like this:
"The integrated UVI in the case of drought..."
"The integrated UVI in the case of floods...."